# Current Medical and Surgical Treatment of Hidradenitis Suppurativa—A Comprehensive Review

**DOI:** 10.3390/jcm11237240

**Published:** 2022-12-06

**Authors:** Lennart Ocker, Nessr Abu Rached, Caroline Seifert, Christina Scheel, Falk G. Bechara

**Affiliations:** International Centre for Hidradenitis Suppurativa/Acne Inversa (ICH), Department of Dermatology, Venereology and Allergology, Ruhr-University Bochum, 44791 Bochum, Germany

**Keywords:** hidradenitis suppurativa, acne inversa, dermatology, inflammation, treatment, adalimumab, targeted therapy, small molecules

## Abstract

Hidradenitis suppurativa (HS) is a chronic inflammatory skin disease presenting with recurrent inflammatory lesions in intertriginous body regions. HS has a pronounced impact on patients’ quality of life and is associated with a variety of comorbidities. Treatment of HS is often complex, requiring an individual approach with medical and surgical treatments available. However, especially in moderate-to-severe HS, there is an urgent need for new treatment approaches. In recent years, increased research has led to the identification of new potential therapeutic targets. This review aims to give a comprehensive and practical overview of current treatment options for HS. Furthermore, the clinically most advanced novel treatment approaches will be discussed.

## 1. Introduction

Hidradenitis suppurativa (HS) is a chronic inflammatory skin disease, presenting with recurrent inflammatory nodules, abscess formation and subsequently formation of subcutaneous sinus tracts and scars during disease progression.

Epidemiologic studies reported varying HS prevalence rates from 0.1% in the US to 1.8% in a Danish population, based on heterogeneous measurement methods [1,2]. A recent meta-analysis came to an overall prevalence of 0.4% [3]. The highest prevalence rates of around 4% are found in young adults between 20 and 40 years, while rates decline in older patients [4].

HS is associated with a high burden of disease, impairing the social and economic situation of affected patients [5]. The occurrence of inflammatory skin lesions and pus discharge in intertriginous areas compromises the sexual life, and causes chronic and exacerbating pain [6,7,8]. HS patients are more often absent from work with subsequent increasing risk of unemployment and loss of social status [9]. Diagnosis is often delayed with a European study reporting a delay of 7.2 years and a recent German cross-sectional study showing more than 10 years and consultation of more than three physicians before correct diagnosis [10,11].

Additionally, HS can be associated with multiple comorbidities [12]. Studies found strong associations with metabolic syndrome, obesity, cardiovascular disease, chronic inflammatory bowel disease, spondylarthritis and depression, underlining the need for a multi-professional treatment approach [12,13,14,15,16,17,18,19,20]. Physicians involved in the treatment of HS patients should be aware of potential comorbidities and further diagnostics should be performed in suspected cases. However, in contrast to other inflammatory skin conditions like psoriasis vulgaris, with adalimumab there is currently only one biologic agent available and the individual selection of treatment modalities is based on multiple factors, like inflammatory activity, objective disease severity and patients’ preferences.

The pathophysiology of HS is complex and has gradually been elucidated in recent years due to increased research interest in inflammatory skin diseases [21,22]. However, we have still recognized only parts of the complex inflammatory mosaic, which is reflected in the lack of efficacy of various treatment approaches in recent trials. Possibly, due to the complex pathophysiology and high inflammatory load, combination therapies might be necessary to control HS on a sufficient level. However, data on the combination of different biologics and/or small molecules are yet missing.

This review aims to provide a comprehensive and clinical-based overview of available treatment options in HS.

## 2. Classification of Disease and Evaluation of Treatment Response

Several classification systems have been described to assess the disease severity of HS. However, a standardized and internationally accepted score is yet missing, resulting in the utilization of different classification tools in clinical studies for the evaluation of treatments. In the following, we briefly describe frequently used scoring systems.

The Hurley staging system was first described in 1989 and due to its simplicity is the most widely used classification system for HS in routine clinical practice [23]. The classification of HS into three stages, mainly based on the presence of sinus tracts and scarring, enables a fast and simple clinical based evaluation. However, the Hurley staging system is not applicable for monitoring of HS, as it represents a static and non-quantitative tool and inflammation activity is not captured [24]. A revised Hurley staging system, that takes account of inflammation activity and subcategorizes Hurley stages I and II into mild, moderate and severe disease, has been described recently; however, clinical application is limited [25].

The modified Sartorius score (mSS) represents a more detailed, open-scaled scoring system, which takes account of the number of involved body regions, the number and types of lesions, and the distance between lesions [26]. Although the mSS is a dynamic system, suitable for treatment monitoring, due to its complexity its use is often time-consuming and especially in severe cases difficult to apply [27].

The hidradenitis suppurativa physician global assessment (HS-PGA) score is a frequently used tool to assess disease severity. HS is classified into six severity grades by counting the numbers of inflammatory nodules, abscesses and sinus tracts [28]. As a dynamic scoring tool HS-PGA can be used for treatment evaluation, although especially in cases of extensive disease with multiple lesions the clinical correlation is limited [29].

The severity assessment of hidradenitis suppurativa (SAHS) score was developed by Hessam et al. and is another clinical based scoring tool, that considers the number of affected body regions, fistulas and other inflammatory lesions as well as HS-related pain and the number of new or flared boils in the last four weeks [30]. The SAHS score enables a dynamic evaluation of HS severity and can also be used for treatment evaluation in clinical practice and studies [31].

The hidradenitis suppurativa clinical response (HiSCR) is defined as a ≥50% reduction in inflammatory lesion count and no increase in abscesses or draining fistulas in HS compared to baseline [32]. It is commonly used as a primary endpoint for treatment evaluation in recent studies and has been shown to be more responsive than HS-PGA [33]. According to the psoriasis area and severity index (PASI), HiSCR_75_- and HiSCR_90_- rates are further developed outcome parameters, assessed in current studies [34].

The international hidradenitis suppurativa severity score system (IHS4) represents a dynamic and easy-to-use scoring tool, and assesses disease severity by counting of inflammatory HS lesions (nodules 1 point, abscesses 2 points, sinus tracts 4 points). A total score of 3 or less signifies mild, 4–10 signifies moderate, and 11 or higher signifies severe disease [35]. Recently, a modified and dichotomous IHS4-55 score has been developed as a potential parameter for the measurement of treatment outcomes [36].

Most HS scoring tools are based solely on physical findings. However, clinical examination can be limited, especially in complex cases with deep tissue involvement. The addition of ultrasound to clinical examination could expand the diagnostic spectrum and enable an objective anatomical based assessment of HS lesions [37]. Moreover, a recent study correlated distinct sonographic parameters with treatment responses in HS patients [38]. Several ultrasound-based scoring systems have been developed for classification of HS lesions; however, clinical application is limited up to now [39,40].

## 3. Treatment

Clinical management of HS is often complex and includes medical and surgical treatments, which are often combined, especially in moderate-to-severe disease [41]. The reduction of symptoms and inflammation activity as well as prevention of formation of chronic HS lesions and scarring represent key therapeutic goals. In current guidelines, an individualized patient-oriented approach, based on the individual subjective impact and objective disease severity, is recommended [20,41].

### 3.1. General Recommendations

#### 3.1.1. Smoking

The link between tobacco smoking and HS has been suggested in several studies. In a retrospective study, 92.2% of HS patients were smokers and clinical remission was more often reported among non-smoking patients [42]. Moreover, Sartorius et al. found lower disease severities, evaluated by the mSS, in non-smokers compared to active smokers [27]. At a molecular level, components of cigarette smoke have been demonstrated to further promote inflammation in HS via inhibition of the already compromised Notch signaling, induction of proinflammatory cytokine expression and causing infundibular epithelia hyperplasia and hypercornification [43,44]. Although clear evidence of tobacco as a trigger of HS has not been found yet, patients should be encouraged to quit smoking [20].

#### 3.1.2. Weight Reduction

Overweight and obesity are considered as frequent comorbidities of HS. In two case–control studies an increment of the likelihood of HS with every BMI unit increase was reported [45]. In another retrospective study, the point prevalence of HS was much higher in an obese study population compared to the general population [46]. This may be explained by changes of the skin microbiome and increased friction of skin folds in obese patients [47,48]. However, evidence for improvement of HS after body weight reduction is limited. A retrospective study reported a significant reduction of inflammatory activity in HS patients undergoing bariatric surgery [46,49]. Moreover, obese HS patients reported lower remission rates than non-obese HS patients [42].

Obese patients should be motivated to reduce their body weight by initiating physical activity and dietary changes [50]. In cases of severe obesity, bariatric surgery may be an option [51].

#### 3.1.3. Psychological Support

Due to its chronic recurrent course with painful lesions and purulent discharge, HS has a pronounced impact on the patient’s life and professional support may be required. Esmann et al. described an increased risk of social isolation due to shame and fear of stigmatization [52]. Quality of life can be severely affected by HS and HS has been observed as one of the most distressing conditions in dermatology [53,54]. HS is associated with an increased risk for distinct psychiatric disorders including depression, anxiety disorders and substance-related abuse [55]. Moreover, a meta-analysis found a more than two-fold increased risk for suicide in HS patients [55]. For this reason, physicians treating HS patients should perform a screening for psychiatric comorbidities and initiate psychiatric referral if necessary.

### 3.2. Medical Therapy

#### 3.2.1. Topical and Intralesional Therapy

In cases of mild-to-moderate disease with limited extent, topical therapy can be considered. Topical clindamycin 1%, applied twice daily on involved areas, is the first-line treatment option for mild-to-moderate disease, especially in absence of deep inflammation or sinus tracts [20,56]. In a small prospective study, topical clindamycin effectively reduced inflammatory lesions compared to placebo [57]. In case of refractory disease after three months of treatment, other treatment options should be considered [20].

Resorchinol is a topical agent with keratinolytic, antipruritic and anti-inflammatory properties and can be applied twice daily on active inflammatory lesions [20]. In a retrospective study with 134 patients, topical resorchinol showed a significant improvement of HiSCR and pain compared to topical clindamycin use [58]. A case series assessing the efficacy of topical resorchinol for mild HS reported significant reduction of inflammatory activity and pain in all treated patients [59].

A current phase II study assesses the efficacy of the topical JAK1/JAK2-inhibitor ruxolitinib (NCT 04414514) [60].

Intralesional injections with glucocorticosteroids as triamcinolonacetonid can be considered for the treatment of solitary acute inflammatory nodules, when rapid reduction of inflammation is desired [61,62]. Clinical response with reduction of inflammation is regularly seen after 48–72 h [20].

#### 3.2.2. Systemic Therapies

##### Zinc Gluconate

Zinc salts (zinc gluconate) show anti-inflammatory effects in HS, probably through inhibition of chemotaxis of neutrophil granulocytes, modulation of cytokine expression and anti-androgen properties [63]. In HS, a high-dose therapy with 90 mg zinc gluconate per day in gradual dose escalation can be considered as a maintenance therapy for limited disease [20]. In several small studies, zinc gluconate showed promising results in mild-to-moderate HS [64,65,66]. As zinc competes with copper in gastrointestinal resorption, long-term use of high doses of zinc may cause hypocupremia and anemia; thus, routine monitoring of copper levels and hemoglobin is recommended [67]. Gastrointestinal discomfort is a frequent reported side effect.

##### Systemic Antibiotic Therapy

Patients with refractory disease under topical therapies or with severe inflammation activity are potential candidates for systemic antibiotic therapy (Table 1). A current study assessed the duration of antibiotic treatments in HS patients and showed that in the majority of oral antibiotic courses the duration of treatment was less than 12 weeks [68].

Recent guideline recommendations recognize tetracyclines as the first-line treatment for more widely spread HS in Hurley stage I/II [20]. Established tetracycline antibiotics include tetracycline, doxycycline and minocycline [69]. A prospective study comparing the efficacy of the different tetracycline antibiotics tetracycline, doxycycline and lymecycline showed a reduction of HSS in all treatment groups with greatest response in the tetracycline group. Moreover, a reduction of pain, formation of new inflammatory lesions and an improvement of quality of life was observed in all treatment groups [70].

An antibiotic combination therapy with clindamycin and rifampicin is recommended as first-line therapy for patients with Hurley stage II and moderate-to-severe disease and as second-line therapy for patients who do not respond on oral tetracycline treatment [20,56]. In an open-label prospective study with 56 patients, the combination of clindamycin and rifampicin showed an overall clinical response with reduction of HSS in 79.6%. HSS_50_ was 37% and complete remission (HSS_100_) was observed in 13%. Side effects were observed in 55.6% of patients with diarrhea, abdominal pain and nausea being the most commonly reported [71]. This data is strengthened by several retrospective trials, which came to similar results; however, the majority of patients with initial complete remission relapsed after discontinuation of treatment [72,73,74]. However, due to the role of rifampicin as a potent inductor of the hepatic CYP system, metabolization of clindamycin is intensified within the combination therapy [75,76]. In one study clindamycin blood levels were decreased by around 90% within two weeks after treatment initiation, raising the question of whether the observed response rates under combination antibiotic therapy may be traced to rifampicin alone [77,78]. More research with prospective randomized trials is needed to address this topic.

For more extensive disease with severe inflammation and widespread distribution of inflammatory lesions some intensified antibiotic treatment regimens have been described; however, up to now evidence is based on case report and retrospective analyses, and these therapies have not been evaluated in prospective trials.

Join-Lambert et al. described a broad-spectrum antibiotic combination therapy consisting of rifampicin, moxifloxacin and metronidazole leading to high rates of complete remissions, defined as clearance of all inflammatory lesions, especially in Hurley stage I/II patients [79].

Ertapenem, another broad-spectrum antibiotic, showed rapid improvement in treatment-refractory cases of HS as a rescue therapy [80]. In a retrospective pilot study with 30 patients, a 6 week course of ertapenem led to a significant and sustained improvement of disease severity, assessed with the Sartorius score [81]. Through its rapid clinical improvement, ertapenem may be initiated in severe HS as neoadjuvant therapy for bridging to surgery or other maintenance therapies [80,82,83].

**Table 1 jcm-11-07240-t001:** Overview of systemic antibiotic regimens in clinical studies.

References	Study Design (Level of Evidence, Oxford Criteria [84])	Treatment Regimen	Efficacy Data/Results
**Tetracyclines**			
Jemec et al., 1996 [85]	randomized double-blind trial (n = 46) (*evidence level: 1b*)	top. clindamycin (0.1%) b.i.d. vs. tetracycline 500 mg b.i.d.	- no significant differences between treatment groups
Vural et al., 2019 [86]	retrospective analysis of different HS treatments (n = 139) (*evidence level: 4*)	doxycycline 100 mg b.i.d. for 3 months	- HiSCR achieved in 60% - doxycycline as preferred treatment regimen in Hurley stage I/II
Jørgensen et al., 2021 [70]	prospective follow-up study (n = 108) (*evidence level: 2b*)	tetracycline 500 mg b.i.d. (n = 32) vs. doxycycline 100 mg b.i.d. (n = 31) vs. lymecycline 300 mg b.i.d. (n = 45) median treatment duration: 3.2 months	- significant clinical improvements (reductions of HSS) in approx. 30%; - no significant differences between treatment groups (greatest HSS reduction observed in tetracycline group)
Armyra et al., 2017 [87]	prospective study/case series (n = 20) (*evidence level: 4*)	minocycline 100 mg q.d. + colchicine 0.5 mg b.i.d. for 3 months followed by colchicine 0.5 mg b.i.d. for 6 months	- significant clinical improvements (HS-PGA and DLQI)
**Clindamycin–Rifampicin Combination Therapy**			
Yao et al., 2021 [71]	prospective open-label study (n = 56) (*evidence level: 4*)	clindamycin 300 mg b.i.d. + rifampicin 300 mg b.i.d. for 10 weeks	- at 6 month follow-up: overall response rate 79.6%; HSS_50_ 37%; HSS_100_ 13% - Side effects 55.6% (esp. gastrointestinal)
van der Zee et al., 2009 [73]	retrospective study (n = 34) (*evidence level: 4*)	clindamycin 600 mg b.i.d. + rifampicin 600 mg b.i.d. varying treatment duration (<10 weeks/>10 weeks)	- overall response rate 82.4%; partial remission 35.3%; total remission 47.1% - side effects 38.2% (esp. gastrointestinal)
Gener et al., 2009 [72]	retrospective study (n = 116) (*evidence level: 4*)	clindamycin 300 mg b.i.d. + rifampicin 600 mg q.d. for 10 weeks	- significant reduction of Sartorius score; complete remission in 11% - significant down-staging (Hurley) and reduction of pain
Mendonça et al., 2006 [74]	retrospective study (n = 14) (*evidence level: 4*)	clindamycin 300 mg b.i.d. + rifampicin 300 mg b.i.d for 10 weeks	- overall response rate 82%; complete remission 66.7%
**Intensified Antibiotic Treatments**			
Join-Lambert et al., 2016 [81]	retrospective study (n = 30) (*evidence level: 4*)	ertapenem 1 g iv. q.d. for 6 weeks followed by antibiotic consolidation treatment (rifampicin, moxifloxacin, metronidazole)	after 6 weeks (ertapenem): - significant reduction of median Sartorius score - clinical remission in 67%/26% in Hurley stage I/II after 6 months (consolidation): - further significant clinical improvement - clinical remission 100%/96%/27% in Hurley stage I/II/III
Join-Lambert et al., 2011 [79]	retrospective study (n = 28) (*evidence level: 4*)	rifampicin 10 mg/Kg q.d. + moxifloxacin 400 mg q.d. + metronidazole 500 mg t.i.d. for 6 weeks *, followed by consolidation with rifampicin + moxifloxacin for ≥6 weeks	- complete remission 57% (esp. in Hurley I/II) - most common side effects: gastrointestinal symptoms 64%; vaginal candidiasis 35% (of female pat.)

* In 14 patients initiation therapy with ceftriaxone 1 g iv. q.d. + metronidazole 500 mg t.i.d. for 2 weeks due to severe inflammation.

##### Hormonal Treatment Approaches

Clinical observations in female HS patients with premenstrual flares and cyclic alterations of inflammation activity led to the suggestion that hormonal alterations may influence the course of HS [88]. Moreover, HS is associated with endocrine disorders such as polycystic ovary syndrome (PCOS) and metabolic syndrome [89,90]. However, the role of hormonal influences on the pathogenesis of HS is still unclear [91]. Most data regarding antiandrogenic treatment approaches for HS are based on retrospective analyses, and case reports and prospective trials are rare (Table 2).

Successful disease control with an antiandrogenic therapy containing ethinylestradiol and cyproteronacetate (CPA) has been described in a case series [92]. In a double-blind controlled cross-over trial comparing two contraceptive regimens containing ethinylestradiol and norgestrel or cyproterone acetate, both treatments produced substantial improvement of disease activity; however, there was no significant difference [93]. In a retrospective study, Kraft et al. compared an antiandrogen treatment approach with antibiotic therapies in 66 female HS patients and found significantly superior response rates, 55% vs. 26%, suggesting that antiandrogen therapy should be considered for all women presenting with HS. Moreover, the authors concluded that female HS patients should be investigated for underlying PCOS and insulin resistance [94].

Spironolactone is a potassium-sparing diuretic with antiandrogen properties due to the inhibition of mineralocorticoid receptors [95]. In several retrospective analyses of female HS patients, spironolactone treatment led to a significant reduction of pain and inflammatory lesions, and improved quality of life [96,97,98].

The antidiabetic agent metformin reduces insulin resistance by improving peripheral insulin sensitivity and may have some antiandrogen properties [83]. In several retrospective trials and case reports, metformin showed promising clinical response rates and thus may contribute to disease control in HS as an adjunctive treatment option [99,100,101].

**Table 2 jcm-11-07240-t002:** Clinical studies evaluating hormonal treatment approaches in HS patients.

References	Study Design (Level of Evidence, Oxford Criteria [84])	Treatment Regimen	Efficacy Data/Results
**Antiandrogen Treatments**			
Kraft et al., 2007 [94]	retrospective chart review patients (n = 64, all female) (*evidence level: 3b*)	various antihormonal treatments (n = 29) - ethinylestradiol 50 µg q.d. + CPA 25 mg q.d. - CPA 25 mg q.d. - Spironolactone 100 mg q.d. - CPA 25 mg q.d. + spironolactone 100 mg q.d.	- antiandrogen therapy superior to oral antibiotics (clinical response 55% vs. 26%) - prevalence of PCOS 38.1%
Sawers et al., 1986 [92]	case series (n = 4, all female) (*evidence level: 4*)	CPA 100 mg q.d. + ethinylestradiol 30–50 mg q.d.	- successful disease control in all patients - reduction of CPA dose led to HS deterioration in 75%
Mortimer et al., 1986 [93]	randomized double-blind crossover trial (n = 24, all female) (*evidence level: 2b*)	- ethinylestradiol 50 µg q.d. + CPA 50 mg q.d. - ethinylestradiol 50 µg q.d. + norgestrel 500 mg q.d.	- substantial improvement of HS in both treatment regimens; no significant difference - antiandrogen therapy may be beneficial in HS treatment
**Spironolactone**			
Lee et al., 2015 [97]	case series (n = 20, all female) (*evidence level: 4*)	spironolactone 100–150 mg/d ± minocycline ± CPA	- clinical improvement (reduction of HS-PGA) in 85% after 3 months; complete remission in 55% - spironolactone was well tolerated and patients were satisfied with treatment
Golbari et al., 2019 [96]	retrospective single-center chart review (n = 67, all female) (*evidence level: 4*)	spironolactone 25–200 mg/d (average dose 75 mg/d)	- significant reduction of HS-PGA, pain and inflammatory lesions - no significant difference in HS improvement between spironolactone ≤75 mg/d and ≥100 mg/d
Quinlan et al., 2020 [98]	retrospective study (n = 26, all female) (*evidence level: 4*)	spironolactone 50–100 mg/d ± metformin	- reduction of DLQI >5 in 35% - no further clinical data
**Metformin**			
Verdolini et al., 2013 [101]	prospective study (n = 25) (*evidence level: 4*)	metformin 1000–1700 mg/d over 24 weeks	- significant clinical improvement (reduction of Sartorius score and number of work days lost) in 72% - significant improvement of DLQI in 64%
Jennings et al., 2020 [99]	retrospective chart review (n = 53) (*evidence level: 4*)	metformin; mean dose 1.5 g/d; mean treatment duration 11.3 months	- subjective clinical response in 68%; complete remission of inflammatory skin lesions in 19% - insulin resistance present in 75%, but not predictive for treatment response
Moussa et al., 2020 [100]	retrospective study (n = 16, pediatric HS patients) (*evidence level: 4*)	metformin as adjunctive treatment (dose not specified)	- clinical improvement in 5/16 (31.3%); no improvement in 5/16 (31.3%); 6/16 patients (37.5%) lost to follow-up

##### Retinoids

Retinoids influence cell differentiation and may have a beneficial impact in HS by reducing keratinocyte proliferation and subsequently preventing the plugging of the pilosebaceous unit (Table 3) [20]. Moreover, retinoids show anti-inflammatory properties by modifying monocyte chemotaxis [102]. The most common adverse events are retinoid dermatitis and, among women, hair loss. Retinoids have a teratogenic effect and effective contraception must be ensured in patients prior to treatment initiation [20].

Acitretin has been evaluated for the treatment of moderate-to-severe HS in a small prospective trial with 17 patients and showed an overall response rate of 47%; however, another 47% of patients dropped out due to lack of efficacy or adverse events [103]. In a retrospective study with 12 patients, acitretin showed promising results with all treated patients achieving clinical remission and reduction of pain [104]. Another recent retrospective cohort study reported significant clinical responses under acitretin treatment and identified the follicular HS phenotype, a history of follicular plugging diseases and a family history of HS as potential predictive markers for treatment response [105].

Isotretinoin has only shown limited efficacy for treatment of HS and several retrospective studies reported treatment response rates between 16.1% and 35.9% [106,107,108]. Clinical exacerbations and occurrence of new flares of HS have been reported after initiation of isotretinoin in various cases, which could be explained by the further reduction of the size and action of sebaceous glands due to isotretinoin therapy [109,110].

**Table 3 jcm-11-07240-t003:** Clinical studies evaluating retinoids for HS treatment.

References	Study Design (Level of Evidence, Oxford Criteria [84])	Treatment Regimen	Efficacy Data/Results
**Acitretin**			
Boer et al., 2011 [104]	retrospective study (n = 12) (*evidence level: 4*)	acitretin monotherapy; mean dose 0.59 mg/Kg/d; duration 9–12 months (mean 10.8 months)	- all patients achieved clinical remission and reduction of pain - 75% showed prolonged clinical responses
Matusiak et al., 2014 [103]	prospective study (n = 17) (*evidence level: 4*)	acitretin; mean dose 0.56 ± 0.08 mg/Kg/d; treatment duration 9 months	- HSSI (≥50% reduction from baseline) reached in 47% - relapse in most patients after discontinuation of acitretin
Tan et al., 2016 [111]	retrospective study (n = 14) (*evidence level: 4*)	acitretin as monotherapy (43%) or adjunctive to other systemic treatments (57%) (dose not specified)	- no patient under acitretin monotherapy achieved clinical response compared to 87.5% in combination group - acitretin ineffective as monotherapy for HS
Sánchez-Díaz et al., 2022 [105]	retrospective cohort study (n = 62) (*evidence level: 4*)	acitretin (dose not specified)	- significant reduction of IHS4 score - follicular HS phenotype and history of follicular plugging as positive predictive markers
**Isotretinoin**			
Boer et al., 1999 [106]	retrospective study (n = 68) (*evidence level: 4*)	low-dose isotretinoin; mean dose 0.56 mg/Kg/d; for 4–6 months	- limited efficacy with clinical response in 23.5% and 16.2% maintained responses during follow-up - isotretinoin more effective in mild HS
Soria et al., 2009 [108]	survey-based retrospective study (n = 87) (*evidence level: 4*)	isotretinoin (dose and treatment duration not specified)	- 16.1% of patients reported clinical improvement; 83.9% reported no improvement or worsening of HS symptoms
Patel et al., 2019 [107]	survey-based retrospective study (n = 39) (*evidence level: 4*)	isotretinoin (dose and treatment duration not specified)	- clinical improvement reported by 35.9%; no response reported by 64.1% - presence of pilonidal cyst associated with clinical response

##### Adalimumab and Other TNFα Inhibitors

Adalimumab, a fully human, IgG1 monoclonal antibody specific for TNFα, is currently the only biologic therapy approved for the treatment of moderate-to-severe HS in adults and adolescent patients ≥12 years (Table 4) [20,22,112]. In two phase III multicenter, double-blind, placebo-controlled studies (PIONEER I and PIONEER II) adalimumab showed a significant effectivity compared to placebo with HiSCR rates of 41.8% and 58.9% (vs. 26.0% and 27.6% in placebo groups), respectively [113]. In open-label extension studies patients under adalimumab therapy were followed up for a minimum of 60 weeks and showed maintained treatment responses with HiSCR rates of 62.5% at week 36 and 52.3% at week 168, respectively [114]. Moreover, an improvement of quality of life, measured with the DLQI score, was observed under adalimumab [114,115].

In a retrospective study by Marzano et al. an inverse correlation between therapeutic delay and clinical response on adalimumab was found, suggesting the concept of a “window of opportunity” and supporting an early initiation of adalimumab [116]. Moreover, the efficacy and safety of adalimumab in patients undergoing wide-excision surgery was investigated in a prospective multicenter phase IV trial. Patients with perioperative use of adalimumab showed significant higher HiSCR rates compared to placebo (48% vs. 34%, *p* = 0.049) and adalimumab treatment was not associated with an increased risk of peri- and postoperative complications, supporting combined treatment approaches in extensive disease [117].

Adalimumab can be administered in subcutaneous injections in two treatment regimens with 40 mg every week or 80 mg every other week after an initial loading dose [20]. The therapy is usually well tolerated with injection side reactions being the most common side effects and large meta-analyses have shown non-significant safety issues compared to placebo [118]. In very rare cases malignancies (esp. lymphomas) have been reported under therapy [119,120].

In recent years, adalimumab biosimilars have become a frequently used alternative to the adalimumab originator agent and currently six adalimumab biosimilars are approved by the FDA and EMA [121]. Several studies have demonstrated the bioequivalence and similar safety profiles of the adalimumab biosimilars compared to the originator agent [122,123]. However, a loss of treatment efficacy has been observed after switching from adalimumab originator to biosimilars in clinical studies [124,125,126]. Kirsten et al. evaluated the clinical responses of 94 patients after switching from adalimumab originator to biosimilar ABP 501, and reported a loss of response or new onset of adverse events in 33.3% [127]. Another retrospective cohort study observed a more rapid loss of efficacy in patients taking adalimumab biosimilars compared to the originator agent and additionally found a greater risk for loss of efficacy in switchers compared to non-switchers, suggesting that, in the management of HS, treatment should begin and continue with the same drug [128]. Prospective studies are needed to further clarify this issue [129].

Infliximab is a chimeric mouse/human IgG1 monoclonal antibody with high affinity to soluble and transmembrane bound TNF⍺. A recent metanalysis calculated a pooled response rate for infliximab of 83% in patients with moderate-to-severe HS and described a low toxicity with a rate of 2.9% for severe adverse events [130]. In a phase II placebo-controlled crossover study, 57% of patients under infliximab treatment reached a reduction of the HS Severity index (HSSI) of >50% compared to 5% in the placebo group. Additionally, improvements in pain intensity and quality of life were observed with concomitant reduction in clinical markers of inflammation [131]. In another prospective trial, sustained treatment responses were observed in HS patients after a single course of infliximab [132]. Overall, infliximab is a promising treatment option for HS; however, larger prospective randomized trials are needed to investigate its efficacy compared to other biologic therapies [130].

Etanercept, a fusion protein consisting of the extracellular ligand-binding domain of the 75 kDa TNF⍺ receptor and the Fc portion of human IgG1, inhibits TNF⍺ signaling through competitive binding of its ligand [133]. In a small prospective placebo-controlled phase II trial, etanercept showed no significant efficacy compared to placebo [134].

Other TNF⍺ inhibitors like golimumab and certolizumab pegol showed inconstant response rates, and evidence is limited on several case reports and small retrospective studies [135,136,137,138,139,140].

**Table 4 jcm-11-07240-t004:** Clinical studies evaluating adalimumab (ADA) for HS treatment.

References	Study Design (Level of Evidence, Oxford Criteria [84])	Treatment Regimen	Efficacy Data/Results
Kimball et al., 2012 [28]	phase II randomized placebo-controlled study (n = 154) (*evidence level: 1b*)	- ADA 40 mg every week from week 4, after initial doses of 160 mg at week 0 and 80 mg at week 2 - ADA 40 mg every other week (EOW) from week 1, after initial dose of 80 mg at week 0 - placebo	- clinical responses (HS-PGA reduction ≥2 from baseline) in 17.6% and 9.6% in ADA groups (weekly/EOW) vs. 3.9% in placebo at week 16 - switching from ADA weekly to ADA EOW dosing resulted in decrease in response
Kimball et al., 2016 [113]	phase III randomized placebo-controlled study (PIONEER I) (n = 307) (*evidence level: 1b*)	- ADA 40 mg every week from week 4, after initial doses of 160 mg at week 0 and 80 mg at week 2 - placebo	- HiSCR achieved in 41.8% vs. 26.0% (ADA vs. placebo) at week 12
Kimball et al., 2016 [113]	phase III randomized placebo-controlled study (PIONEER II) (n = 326) (*evidence level: 1b*)	- ADA 40 mg every week from week 4, after initial doses of 160 mg at week 0 and 80 mg at week 2 ± tetracycline antibiotics - placebo ± tetracycline antibiotics	- HiSCR achieved in 58.9% vs. 27.6% (ADA vs. placebo) at week 12 - significant reduction of inflammatory lesions, pain and modified Sartorius score
Zouboulis et al., 2019 [114]	phase III open-label extension study of PIONEER I/II (n = 88) (*evidence level: 2b*)	ADA 40 mg every week	- HiSCR achieved in 52.3% at week 168 - ADA can be considered for long-term control of HS
Marzano et al., 2021 [116]	retrospective cohort study (n = 389) (*evidence level: 2b*)	ADA 40 mg every week from week 4, after initial doses of 160 mg at week 0 and 80 mg at week 2	- HiSCR achieved in 43.7% at week 16 and in 53.9% at week 52 - significant reduction of DLQI and pain - inverse correlation between therapeutic delay and response to ADA
Bechara et al., 2021 [117]	phase IV randomized placebo-controlled study (n = 206) (*evidence level: 1b*)	HS patients undergoing wide-excision surgery - ADA 40 mg every week - placebo	- HiSCR achieved in 48% vs. 34% (ADA vs. placebo) at week 12 - continuation of ADA treatment was not associated with increased risk for peri-/postoperative complications

### 3.3. What’s Coming Next? Promising Molecular Targets for Future HS Treatment

The increasing interest in inflammation research in recent years contributed to a deeper understanding of the pathogenesis of HS and led to the identification of potential molecular targets for new treatment options [141]. Due to capacity reasons this review focusses on the most advanced clinical approaches, namely targeting the interleukin-17, interleukin-1 axis and the JAK/STAT signaling (Table 5).

#### 3.3.1. Interleukin-17

In recent years, interleukin 17 (IL-17) has emerged as a major player in various autoimmune and inflammatory skin disorders, and its proinflammatory isoforms IL-17A, IL-17C and IL-17F are presumed as key cytokines in HS [142,143]. IL-17A and IL-17F are mainly produced by T-helper 17 cells (T_H17_) and play a major role in control of many fungal and bacterial infections, while IL-17C is released by epithelial cells like keratinocytes, and promotes further inflammation by mediation of the production of other inflammatory molecules and further enhancing the inflammation cascade through stimulation of IL-17A and IL-17F production in T_H17_-cells as a feed forward loop [144,145].

The first evidence for the role of IL-17 in HS was reported by Schlapbach et al., who found a 30-fold increased expression of IL-17A in HS lesions compared to healthy skin [146]. Another study confirmed these findings and identified a dysregulated cytokine milieu in perilesional skin, suggesting that subclinical inflammation may be present in HS skin prior to the formation of active lesions [147]. Moreover, Navrazhina et al. found significantly increased expressions of the proinflammatory IL-17 isoforms A, C and F in lesional, perilesional and even unaffected skin of HS patients compared to healthy individuals [148]. Given these accumulating findings pointing toward a key role of IL-17 signaling in the pathogenesis of HS, targeting the IL-17 pathway seems like a promising treatment approach. However, as HS can be associated with inflammatory bowel disease, it should be kept in mind that paradoxical exacerbations of pre-existing IBD have been reported under anti-IL17 treatment [90,149].

Secukinumab is a recombinant human monoclonal IgG1κ antibody that selectively targets IL-17A and blocks its interaction with the IL-17 receptor [150]. In an open-label pilot study with nine patients, 78% reached HiSCR after 24 weeks of treatment with secukinumab 300 mg every 4 weeks [151]. Another open-label trial with 20 enrolled patients tested two dose levels of secukinumab (300 mg every 2 or 4 weeks after an initial loading dose) and observed pooled HiSCR rates of 70%. Interestingly, clinical responses were also observed in patients with failure to prior anti-TNF⍺ treatment [152].

Recently, the first results of two randomized, placebo-controlled, multicenter phase III trials (SUNSHINE, SUNRISE) evaluating the efficacy of secukinumab in two dose regimens (every 2 weeks or every 4 weeks) have been presented. After 16 weeks, the primary endpoint was reached in both studies for the every 2 weeks (Q2W) regimen and in one for the every 4 weeks (Q4W) regimen, demonstrating the superiority of secukinumab over placebo in patients with moderate-to-severe HS (HiSCR rates: 45% vs. 33.7% (study 1) and 42.3% vs. 31.2% (study 2) for secukinumab 300 mg every 2 weeks, and 41.8% vs. 33.7% (study 1) and 46.1% vs. 31.2% (study 2) for secukinumab every 4 weeks). In both treatment groups secukinumab significantly reduced inflammatory lesions and pain, and improved patients’ quality of life [153].

Bimekizumab is another humanized IgG1κ monoclonal antibody, targeting IL-17A and IL-17F. In a recent double-blind, placebo-controlled, phase II clinical trial (ClinicalTrials.gov NCT03248531) 90 patients were randomized to receive bimekizumab (640 mg at week 0, 320 mg every 2 weeks), adalimumab (160 mg at week 0, 80 mg at week 2 and 40 mg every week for weeks 4–10) or placebo. HiSCR was reached in 57.3% of the bimekizumab group compared to 26.1% in the placebo arm. Moreover, 46% of the patients under bimekizumab achieved HiSCR_75_ and 32% achieved HiSCR_90_, compared to 10% and 0% in the placebo group, respectively [34]. These promising results led to the initiation of three placebo-controlled phase III studies, which are currently ongoing (NCT04901195, NCT04242498, NCT04242446).

Nanobodies represent a novel innovative class of antibody-derived targeted therapies. Consisting of one or more domains based on the small antigen binding variable regions of heavy chain antibodies, they are much smaller compared to conventional monoclonal antibodies, facilitating their tissue penetration [154]. Moreover, several nanobodies can be linked to obtain multi-specific molecules [154,155].

Sonelokimab is a trivalent nanobody containing three domains with specificity for IL-17A, IL-17F and human serum albumin, and has already showed clinical efficacy in the treatment of plaque psoriasis in a recent phase II study [156]. An ongoing randomized placebo-controlled trial evaluates the efficacy of sonelokimab in patients with moderate-to-severe hidradenitis suppurativa (ClinicalTrials.gov NCT05322473) [157].

Izokibep, another small, molecular, antibody-mimetic IL-17A inhibitor has recently shown significant clinical responses in patients with psoriatic arthritis and is currently in a placebo-controlled phase II trial in HS (ClinicalTrials.gov NCT05355805) [158].

#### 3.3.2. Interleukin-1

The proinflammatory interleukin-1 pathway is involved in promoting inflammation in HS and IL-1β has been shown to be significantly overexpressed in lesional skin of HS patients [159]. As the primary circulating isoform, IL-1β is expressed by macrophages, monocytes and dendritic cells due to activation of pattern recognition receptors in an inflammasome-dependent process [160]. IL-1β induces the expression of molecules promoting remodeling of the extracellular matrix and immune cell infiltration, making it a potential target for future HS treatments [159].

Anakinra is a recombinant IL-1 receptor antagonist, preventing its interaction with the ligands IL-1⍺ and IL-1β [161]. In a phase II placebo-controlled trial, HiSCR rates of 78% have been described after 12 week treatment compared to 30% in the placebo group; however, the majority of patients relapsed after treatment discontinuation [162]. Another prospective open-label study with five patients reported clinical improvements of inflammation activity and quality of life after an 8 week course of anakinra [163]. Several case reports described inconstant clinical responses with partial improvements and failure of anakinra treatment [140,164,165,166,167].

Bermekimab (also known as MABp1) is a recombinant monoclonal antibody that neutralizes IL-1⍺. In a prospective randomized controlled phase II trial, patients with moderate-to-severe HS, that were not eligible for or failed a prior anti-TNF⍺ treatment, showed significant clinical responses with HiSCR rates of 60% compared to 10% in the placebo group after 12 weeks [168]. Patients who were initially randomized to the placebo group were allowed to continue in an open-label extension study and 75% showed clinical response (HiSCR) after 12 weeks of bermekimab treatment [169].

In another phase II study with 42 patients with moderate-to-severe HS who were naïve to or had failed prior anti-TNF⍺ therapy, significant clinical responses were observed in both groups with HiSCR rates of 63% in the TNF⍺-failure group and 61% in the TNF⍺-naïve group, respectively, qualifying bermekimab as potential alternative treatment option for non-responders to anti-TNF⍺ [170].

A randomized placebo and active-comparator-controlled phase IIa/b trial evaluating the efficacy of bermekimab compared to placebo and adalimumab is currently ongoing (ClinicalTrials.gov NCT04988308) [171].

Canakinumab is a human monoclonal antibody targeting IL-1β. There is limited evidence for its efficacy in HS and case reports described contradictory responses [172,173,174].

Lutikizumab, a human dual variable-domain antibody that selectively binds and inhibits IL-1⍺ and IL-1β, has been previously tested as an anti-inflammatory treatment in osteoarthritis [175,176]. A recent phase II trial studies the efficacy of lutikizumab in HS patients with failure to prior anti-TNF⍺ therapy (ClinicalTrials.gov NCT05139602) [177].

#### 3.3.3. JAK/STAT Inhibitors

The Janus kinase and signal transducers and activators of transcription (JAK/STAT) pathway represents a rapid membrane-to-nucleus cell signaling module, which regulates the expression of various critical mediators of cancer and inflammation [178]. Inhibitors of the JAK/STAT pathway are a potential approach for future treatment of HS, as they enable the simultaneous modulation of expression of different cytokines that are involved in the complex inflammatory process of HS [141,179].

The efficacy of the JAK1 inhibitor INCB054707 was evaluated in two phase II studies (ClinicalTrials.gov NCT03569371, NCT03607487) in patients with moderate-to-severe HS. INCB054707 was well tolerated and clinical improvement of disease was observed; however, the effect was only significant in the high-dose group with 90 mg (HiSCR 88% vs. 57% placebo) [180].

Upadacitinib is a second-generation JAK inhibitor with selectivity for JAK1. In a retrospective cohort study 75% and 100% of patients treated with upadacitinib reached HiSCR after 4 and 12 weeks of treatment, respectively. HiSCR_75_ rates were 30% and 95% after 4 weeks and 12 weeks [181]. These promising observations led to the initiation of a phase II randomized controlled trial with results unpublished (ClinicalTrials.gov NCT04430855) [182].

A phase II trial for topical treatment of early-stage HS lesions with the JAK1/JAK2 inhibitor ruxolitinib is currently ongoing (ClinicalTrials.gov NCT04414514) [60].

**Table 5 jcm-11-07240-t005:** Results of clinical studies evaluating new treatment approaches for HS (only completed studies with reported study results).

References	Study Design(Level of Evidence, Oxford Criteria [84])	Treatment Regimen	Efficacy Data/Results
**IL-17 (Secukinumab)**			
Prussick et al., 2019 [151]	open-label pilot study (n = 9) *evidence level: 4*)	secukinumab 300 mg weekly for 5 weeks (loading dose) followed by 300 mg every 4 weeks for 24 weeks	- HiSCR achieved in 78% after 24 weeks - improvement of Sartorius score and DLQI
Casseres et al., 2020 [152]	open-label study (n = 20) (*evidence level: 4*)	secukinumab 300 mg weekly for 5 weeks (loading dose) followed by 300 mg every 2 or 4 weeks	- pooled HiSCR rate of 70% after 24 weeks - clinical responses were also observed in patients with failure to prior anti-TNF⍺ treatment
Reguiaï et al., 2020 [183]	retrospective study (n = 20) (*evidence level: 4*)	secukinumab 300 mg weekly for 5 weeks (loading dose) followed by 300 mg every 4 weeks	- HiSCR achieved in 75% after 16 weeks - maintained clinical responses during follow-up - 2 patients developed Crohn disease after 3 and 5 months of treatment
Ribero et al., 2021 [184]	retrospective multicenter study (n = 31) (*evidence level: 4*)	secukinumab 300 mg weekly for 5 weeks (loading dose) followed by 300 mg every 4 weeks	- HiSCR achieved in 41% after 28 weeks
Kimball et al., 2022 [153]	2 phase III randomized placebo-controlled trials (n = 1084) - NCT03713619 (SUNSHINE) - NCT03713632 (SUNRISE) (*evidence level: 1b*)	- secukinumab 300 mg every 2 weeks (Q2W) - secukinumab 300 mg every 4 weeks (Q4W) - placebo	- HiSCR rates at week 16: SUNSHINE: 45.0% (Q2W); 41.8% (QW4); 33.7% (placebo) SUNRISE: 42.3% (Q2W); 46.1% (Q4W); 31.2% (placebo) - superiority of secukinumab over placebo in moderate-to-severe HS - rapid clinical responses (observed from week 2) - acceptable safety profile
**IL-17 (Bimekizumab)**			
Glatt et al., 2021 [34]	phase II randomized placebo- and active-comparator-controlled trial (n = 90) (*evidence level: 1b*)	- bimekizumab 640 mg at week 0 followed by 320 mg every 2 weeks - adalimumab 160 mg at week 0, 80 mg at week 2, followed by 40 mg every week - placebo	- HiSCR was achieved in 57.3% in the bimekizumab group at week 12 (vs. 26.1% in placebo group) - HiSCR75 rates of 46.0% in bimekizumab group at week 12 (vs. 10% in placebo group; 35% in adalimumab group) - HiSCR90 reached in 32% of patients with bimekizumab (vs. 0% in placebo group; 15% in adalimumab group)
**IL-1 (Anakinra)**			
Leslie et al., 2014 [163]	open-label study (n = 6) (*evidence level: 4*)	anakinra 100 mg daily for 8 weeks	- clinical responses with reduction of mSS and improvement of DLQI - rapid relapses after treatment discontinuation in the follow-up period
Tzanetakou et al., 2016 [162]	phase II randomized placebo-controlled trial (n = 20) (*evidence level: 2b*)	- anakinra 100 mg daily for 12 weeks - placebo	- HiSCR achieved in 78% after 12 weeks (vs. 30% in placebo group) - modulation of cytokine production in peripheral blood mononuclear cells - majority of patients relapse after treatment discontinuation
**IL-1** **⍺** **(MABp1/Bermekimab)**			
Kanni et al., 2018 [168]	phase II randomized placebo-controlled trial (n = 20); patients not eligible for anti-TNF⍺ treatment (*evidence level: 2b*)	- MABp1 (bermekimab) 7.5 mg/Kg every 2 weeks for 12 weeks - placebo	- HiSCR achieved in 60% vs. 10% (MABp1 vs. placebo) at week 12 - maintained clinical responses in 40% after 24 weeks
Kanni et al., 2021 [169]	open-label extension study of NCT02643654 (n = 8 *^1^) (*evidence level: 4*)	MABp1 (bermekimab) 7.5 mg/Kg every 2 weeks for 12 weeks	patients initially randomized to placebo achieved HiSCR in 75% after 12 weeks of MABp1 treatment
Gottlieb et al., 2020 [170]	phase II open-label study (n = 42) (*evidence level: 3b*)	bermekimab 400 mg every week for 12 weeks in - patients with failure to prior anti-TNF⍺ therapy (group A) - patients naïve to anti-TNF⍺ therapy (group B)	- comparable HiSCR rates after 12 weeks with 63% (group A) and 61% (group B) - bermekimab as potential treatment alternative for TNF⍺ non-responders
**JAK/STAT Inhibitors**			
Alavi et al., 2022 [180]	open-label study (NCT03569371) (n = 10) (*evidence level: 4*)	INCB054707 15 mg p.o. daily for 8 weeks	- HiSCR achieved in 43% at week 8 - 70% experienced at least one AE (esp. upper respiratory tract infections)
Alavi et al., 2022 [180]	phase II randomized placebo-controlled trial (NCT03607487) (n = 35) (*evidence level: 2b*)	- INCB054707 in escalating dose regimens (30/60/90 mg daily) for 8 weeks - placebo	- dose dependent clinical responses; HiSCR rates of 56%/56%/88% (30/60/90 mg group) vs. 57% (placebo) - asymptomatic thrombocytopenia occurred in 4 patients of the 90 mg group, resolved after dose interruption
Kozera et al., 2022 [181]	retrospective cohort study (n = 20) (*evidence level: 4*)	upadacitinib for 12 weeks - 15 mg p.o. daily until week 4 - at week 4: patients failing HiSCR were switched to 30 mg daily	- HiSCR rates of 75% after 4 weeks and 100% after 12 weeks - HiSCR75 achieved in 95% after 12 weeks

*^1^ Patients initially randomized to placebo group.

### 3.4. Surgical Treatment Options

There are various options for surgical intervention, requiring an individualized strategy that considers multiple factors such as the types of lesions, inflammation activity and patients’ preferences [185,186]. In general, surgical intervention in HS is mostly reserved for advanced cases with irreversible tissue destruction, such as sinus tracts, scars and contractions [187]. Malignancies, especially cutaneous squamous cell carcinoma, represent a rare, yet severe complication in HS patients and, in suspected cases, an early surgical excision is recommended [188]. Moreover, surgery can be considered for chronic inflammatory lesions that do not respond on conservative therapies and for the treatment of acute inflammatory lesions when rapid symptom relief is desired [20]. However, it should be noted that there is no generally accepted consensus on the definition and application of distinct surgical techniques in HS and comparative studies are rare [22]. Moreover, a clear definition of the terminus “recurrence after surgery” is still missing.

#### 3.4.1. Incision and Drainage

In acute cases with abscess formation, incision and drainage can be considered for acute pain relief, although this is only a symptom treatment with nearly 100% recurrence rates [189]. As deroofing can be performed approximately in the same amount of time, coming with higher recurrence-free rates, experts recommend deroofing over incision and drainage [41].

#### 3.4.2. Deroofing

Deroofing describes a superficial removal of the skin covering an inflammatory nodule or a solitary sinus tract, exposing the partially epithelialized basis of the lesion with subsequently curettage of the gelatinous granulation tissue [190,191,192]. The advantages of this procedure are its simplicity, cost-effectiveness and that it can be performed under local anesthesia, qualifying this method for treatment of inflammatory nodules, abscesses and solitary sinus tracts [186]. In a small study with 44 patients undergoing deroofing of axillary or inguinal HS lesions, 87% were recurrence-free in the follow-up interval of five years [193].

The skin-tissue-sparing excision with electrosurgical peeling (STEEP) method represents a similar surgical approach and was first described by Blok et al. [194]. Here, successive tangential excisions are performed with an electrosurgical wire loop until the epithelialized bottom of the lesion is exposed, whilst saving as much healthy tissue as possible. Although this surgical technique is associated with a short time to wound healing and a low risk of wound contraction, no long-term outcomes are reported and evidence is limited on small case series [195].

#### 3.4.3. Excision

Excision describes the complete removal of the affected tissue and represents a more invasive surgical approach in HS. Depending on the extent of resection experts differentiate between limited, wide and radical excisions; however, there are no generally accepted definitions and distinctions between surgical interventions are blurred [186]. Van Rappard et al. defined limited or localized excisions as the complete excision of the affected tissue, beyond the borders of activity, leaving clear margins [196]. Limited excisions represent a low-invasive surgical approach, appropriate for the treatment of recurrent inflammatory nodules or abscesses and solitary sinus tracts in Hurley stage I or II, and can be performed under local anesthesia in an outpatient setting [196]. Wide excisions are understood as the surgical removal of the affected tissue including the surrounding subcutaneous fat and perilesional skin with an additional resection margin [197,198]. In severe cases, radical excisions with the complete removal of the entire hair bearing area including the subcutaneous fat to the underlying fascia of an affected body region can be performed [20]. Nesmith et al. suggested an additional superficial lymphadenectomy for a further reduction of the risk of recurrence [199].

In cases of extensive disease with multiple or confluent sinus tracts, preoperative MRI- or ultrasound-based imaging methods or an intraoperative use of dye mapping methods, such as methyl violet or iodine starch, can contribute to an improved visualization of the surgical site [200,201,202].

Although there are no accepted definitions for surgical techniques and recurrence in HS, it is generally considered that more extensive resections are associated with a lower risk of recurrence [203].

#### 3.4.4. Laser-Based Therapies

Various laser and light-based therapies have gained increasing attention as a possible treatment modality for HS [204].

The carbon dioxide (CO_2_) laser acts at a wavelength of 10,600 nm and can be used for tissue ablation in different modalities, namely vaporization and excision. CO_2_-laser vaporization represents a tissue-sparing treatment option, that aims on a focal destruction of chronic or inflammatory HS lesions, and was first described by Dalrymple et al. for the treatment of sinus tracts [205]. In two case series, Lapins et al. reported low local recurrence rates and satisfactory postoperative outcomes after a stepwise horizontal CO_2_-laser evaporation of sinus tracts in Hurley stage II patients [206,207]. Another retrospective study showed similar outcomes, but described local recurrence rates of 29% [208]. A retrospective study evaluated the outcomes of 185 treated sites of 61 HS patients after CO2-laser excision or marsupialization with secondary intention healing, and demonstrated fast healing rates and a local recurrence rate of 1% [209].

Summarizing, CO_2_-laser interventions are tissue-sparing treatment options for HS lesions with fast postoperative wound healing and low complication rates. As they can be performed under local anesthesia and wounds are generally allowed to heal by secondary intention, these treatments seem suitable for an outpatient setting [204].

Neodymium-doped yttrium aluminum garnet (Nd:YAG) lasers act as non-ablative lasers at a wavelength of 1064 nm and enable a selective photothermolysis of hair follicles [210]. Their efficacy has been demonstrated in several randomized controlled studies, leading to significant clinical improvements and prevention of new inflammatory lesions in the treated body areas of HS patients [211,212]. In a histopathologic study, an initially increased perifollicular inflammation, observed after 1 week of treatment, was followed by a fibrotic tissue transformation after 4–8 weeks of treatment [213]. Axillary and inguinal areas have been shown to be more responsive to Nd:YAG-laser treatment compared to inframammary and gluteal regions [212].

Moreover, combination treatments consisting of Nd:YAG-laser-mediated hair follicle destruction and CO_2_-laser ablation of HS lesions have shown superior clinical responses [214,215].

#### 3.4.5. Postoperative Wound Management and Wound Closure Options

There are various options for wound closure after surgical intervention, including healing by secondary intention, application of skin grafts and primary wound closure techniques [20]. Rompel et al. analyzed the outcomes of 106 HS patients undergoing radical excision and found no correlation between recurrence rates and the type of surgical reconstruction [201]. However, a current meta-analysis reported recurrence rates following wide surgical excision with 15% after primary wound closure, 8% after skin flap reconstruction and 6% after application of skin grafts, respectively [216]. Ovadja et al. described similar results with the highest recurrence rates after primary wound closure and lowest recurrence rates after skin graft coverage [217].

Negative pressure wound therapy (NPWT) has been shown to improve wound healing and wound granulation by increasing tissue oxygenation and reducing the bacterial load [218]. In HS, NPWT can be considered for postoperative wound management following wide or radical excisions. After completion of NPWT-assisted wound granulation, healing by secondary intention or defect coverage with skin grafts can be performed [219]. A small study analyzing NPWT for axillary wound management following wide excisions in HS patients reported a reduced time of hospitalization, and low recurrence and complication rates [220].

Postoperative secondary intention healing is a well established approach for wounds after surgical intervention with low recurrence rates and favorable functional outcomes [221]. However, the time to complete wound healing is generally longer than in primary wound closure options and frequent follow-up visits should be recommended to ensure optimal results [222].

Skin grafts can be considered for defect coverage following extensive excisions or in critical areas with a high risk for postoperative scar contraction. In HS split-thickness skin grafts (STSG) are the most frequently used skin grafts, as in comparison to full thickness skin grafts harvesting is easier and less invasive [223]. If required, meshing of STSG can achieve coverage of larger wound areas [224]. In a comparative study by Morgan et al. split-thickness skin grafts showed faster wound healing than healing by secondary intention. However, secondary healing with frequent wound dressing resulted in certain healing with good cosmetic results and avoided the need for immobilization and a painful donor site. Most patients in this study preferred secondary wound healing to skin grafting [222]. Another retrospective study analyzed the outcomes of HS patients after axillary reconstruction with STSG and skin flaps, and found no significant differences in postoperative outcome and recurrence rates [225].

Primary wound closure represents an option for closure of small excision wounds, especially in cases with limited disease [196]. However, primary wound closure is associated with a high risk of local recurrence [216,226].

Skin flaps can be considered as a more complex reconstructive approach for wounds after larger excisions, especially when critical anatomical structures, such as nerves or blood vessels, are exposed or when there is a risk for postoperative scarring and contraction [227]. As local flaps bear the risk of containing HS affected skin, reconstruction with skin flaps should only be performed after wide surgical excisions [228]. Currently, there is a large number of different skin flaps that have been described for HS reconstruction and frequently used flaps are lateral thoracic island fasciocutaneous flaps [229], Limberg transposition flaps [230], fasciocutaneous perforator-based flaps [231], myocutaneous flaps [232,233] and thoracodorsal artery perforator flaps [234,235].

## 4. Conclusions

Treatment of hidradenitis suppurativa is often challenging and should be performed with an individualized, patient-oriented approach considering medical and surgical treatment modalities. Several promising novel treatment approaches, including antibodies, nanobodies, small molecules and hormonal treatments, are currently being investigated in clinical studies and could enrich the therapeutic spectrum for HS.

## Data Availability

Not applicable.

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
