# Peer review of "Current Medical and Surgical Treatment of Hidradenitis Suppurativa—A Comprehensive Review"

_jcm, 2022, doi:10.3390/jcm11237240_

Round 1
Reviewer 1 Report
The authors conducted a very interesting review in which they analyzed all available treatments for hidradenitis suppurativa.The review although well written and very detailed in my opinion lacks very important information that can be added to complete the collection.
The authors in the treatment section in particular when talking about the role of intralesional therapy may refer to the role of intralesional corticosteroids , the authors might read this article where the authors discuss it: doi: 10.1159/000521712.
It would be very interesting if the authors also discussed the role of new biologics that seem promising, they could add a paragraph as future directions where they can talk about Burzi L, Repetto F, Ramondetta A, Rozzo G, Licciardello M, Ribero S, Quaglino P, Dapavo P. Guselkumab in the treatment of severe hidradenitis suppurativa, a promising role? Dermatol Ther. 2021 May;34(3):e14930. doi: 10.1111/dth.14930. Epub 2021 Mar 14. PMID: 33665949. doi: 10.1016/j.jaad.2019.08.089. Casseres RG, Prussick L, Zancanaro P, Rothstein B, Joshipura D, Saraiya A, Turkowski Y, Au SC, Alomran A, Abdat R, Abudu M, Kachuk C, Dumont N, Gottlieb AB, Rosmarin D. Secukinumab in the treatment of moderate to severe hidradenitis suppurativa: Results of an open-label trial. J Am Acad Dermatol. 2020 Jun;82(6):1524-1526. doi: 10.1016/j.jaad.2020.02.005. Epub 2020 Feb 7. PMID: 32044410. doi: 10.1111/ced.15291.
I pointed out these articles from which the authors can take a cue to add this paragraph that would be very useful for the literature
Reviewer 2 Report
Dear Authors,
thank you for the priviledge to review your paper. The management of HS has been described in a exhaustive manner.
I have only few concerns about the introductory part:
- in the classification / staging section, I think it would be useful to spend some words about the role of ultrasound in the diagnosis, staging (SOS-HS by Wortsman, Dermatol Surg 2013 and US HS-PGA by Nazzaro, Ital J Dermatol Venerol 2021) and follow-up. In particular evaluation of fibrosis with US and vascularization evaluated with Color Doppler demonstrated to be related to the treatment response (Nazzaro et al, Dermatol Ther 2021)
- the recent IHS4-55 dichotomous IHS4 version, based on a 55% reduction of the total score, should be cited (Tzellos T, JEADV 2022)
Reviewer 3 Report
Summary:
The authors performed a review regarding “management of hidradenitis suppurativa.” The article is well written and comprehensive. However, many review articles on hidradenitis suppurativa have been published in the past 1-2 years, and there are still some points need to be clarified.
Major concern:
1. It is recommended to add the section of pathogenesis of hidradenitis suppurativa so that readers can easily understand the mechanism of drug therapies.
2. Hidradenitis suppurativa is associated many comorbidities, such as metabolic syndrome, cardiovascular disease, inflammatory bowel disease, spondylarthritis and depression. Did the treatment of hidradenitis suppurativa affect these comorbidities? Besides, how do you recommend clinicians screen these possible comorbidities in patients with hidradenitis suppurativa?
Minor concern:
1. It is recommended to add the evidence level of clinical studies in Table 1-4.
2. Consider adding table 5 to demonstrate the summary of clinical studies on interleukin-17, interleukin 1, and JAK/STAT-inhibitors.
Round 2
Reviewer 3 Report
The authors have adequately responded to reviewers' comments.